# Developing an Automated Analytical Process for Disaster Response and Recovery in Communities Prone to Isolation

**DOI:** 10.3390/ijerph192113995

**Published:** 2022-10-27

**Authors:** Byungyun Yang, Minjun Kim, Changkyu Lee, Suyeon Hwang, Jinmu Choi

**Affiliations:** 1Department of Geography Education, Dongguk University, Seoul 04620, Korea; 2Department of Geography, Kyung Hee University, Seoul 02447, Korea

**Keywords:** communities at risk of isolation, evacuation routes, disaster management, web map

## Abstract

Today, unpredictable damage can result from extreme weather such as heat waves and floods. This damage makes communities that cannot respond quickly to disasters more vulnerable than cities. Thus, people living in such communities can easily become isolated, which can cause unavoidable loss of life or property. In the meantime, many disaster management studies have been conducted, but studies on effective disaster response for areas surrounded by mountains or with weak transportation infrastructure are very rare. To fill the gap, this research aimed at developing an automated analysis tool that can be directly used for disaster response and recovery by identifying in real time the communities at risk of isolation using a web-based geographic information system (GIS) application. We first developed an algorithm to automatically detect communities at risk of isolation due to disaster. Next, we developed an analytics module to identify buildings and populations within the communities and efficiently place at-risk residents in shelters. In sum, the analysis tool developed in this study can be used to support disaster response decisions regarding, for example, rescue activities and supply of materials by accurately detecting isolated areas when a disaster occurs in a mountainous area where communication and transportation infrastructure is lacking.

## 1. Introduction

### 1.1. Background

As heavy rains, typhoons, hurricanes, and tornadoes continue to occur every year, natural disasters such as floods and wildfires also frequently ensue, causing great loss of human life and destruction of the natural environment [1,2]. In particular, in areas having an hourly rainfall of 100 mm or more, landslides and flooding are common, and this leads to frequent bridge collapse and road damage [3,4]. Moreover, areas surrounded by mountains or prone to flooding are likely to be isolated from assistance in the event of a disaster [5]. Therefore, it is essential for disaster managers to have a strategy to prepare for or recover from disasters in highly vulnerable communities where isolation can easily occur.

To prepare for an actual disaster, two preliminary steps are needed to help people living in areas that are likely to be isolated [6]: (1) the areas that are likely to become isolated must be detected in advance and (2) information on evacuation routes and shelters for residents in the isolated areas must be identified [7]. This information should be easily accessible in real time through a mobile environment. In addition, if disaster managers can estimate the number of people living in such at-risk areas, this can be used as valuable data for rapid lifesaving activities [8]. For this reason, it is necessary to develop a disaster management system that can be used from the time immediately after a disaster until the recovery stage by precisely performing an analysis of the area at risk of isolation in advance. Furthermore, this system should be easy for residents or disaster managers to use. If such preparatory work can be systematized by an automated system, it will be possible for disaster managers to prevent any delay in rescue activities and the prolonged isolation of residents in actual disaster situations.

Many disaster management studies have been conducted since the first Earth observation satellite, Landsat 1, was launched in 1972 [9]. Geographic information system (GIS) and remote sensing (RS) technologies have contributed considerably to disaster management, especially to research on the disaster management cycle, which has four stages: preparation, response, recovery, and mitigation [10,11,12,13]. To date, various studies have been conducted in the RS and GIS fields to support disaster management, including an analysis of areas flooded by storm surge or heavy rain [14,15,16], an analysis to automatically identify areas of mountains destroyed by wildfires [17,18], a time-series change detection analysis of damage to natural or manmade features [19,20], and a study related to the development of advanced machine or deep learning methods [21,22]. Most of these studies have focused on places or spaces directly affected by disasters rather than on isolated communities affected by disasters.

Despite the progress in this area, there have been few disaster-related studies on local communities prone to isolation. Thus, this study poses the following research questions.
(1)What are the essential geospatial technologies that can be used to help people living in communities that are vulnerable in the event of a disaster?(2)Why do we need an analytical process to identify and analyze isolated communities in real disaster situations, and what analytical procedures are required for the communities that are prone to isolation?(3)What kind of systems can disaster managers use to deliver information relevant in isolated areas to citizens in real time, and how can they efficiently operate them?(4)Can we develop scenario-based approaches on web maps to rapidly help isolated communities at risk of disasters?


To answer the proposed questions, the purpose of this study is to develop an automated analysis tool that can be directly used in disaster response and recovery by identifying communities at risk of isolation in real time from a web-based GIS application. First, we develop an algorithm to detect areas isolated by various disaster situations and visualize this data through mapping. In this step, scenario-based approaches are carried out for mountainous and rural areas where disaster response is difficult. Second, analysis is performed by selecting case areas that have experienced damage from recent disasters. In the analytical phases, the proposed algorithm identifies the number of buildings and the number of people within an isolated area, and it provides optimal evacuation routes for refugees. Moreover, it considers the number of refugees and ensures that they are efficiently distributed to nearby shelters. Lastly, a web-based GIS application is developed so that the previously developed analysis processes can be utilized in a web-based environment. This web-based GIS environment is intended for use by refugees and disaster managers for rapid disaster response and recovery.

To achieve the research objectives, this paper is structured as follows: Section 1.2 describes previous disaster management studies that are related to isolated risk areas that have been affected by disaster. Next, Section 2 addresses the study area and materials, including data collection, together with the methods used to develop the algorithms in this study and major analytical procedures. Section 3 subsequently provides the final maps resulting from the algorithms and an interpretation of the outcomes. Finally, Section 4 discusses the significant contributions and limitations of this research, and Section 5 concludes the paper.

### 1.2. Previous Studies

Most of the existing research on isolated communities is focused on responding before and after disasters: for example, research related to making appropriate plans during the disaster preparation phases [23,24], a study on new road construction to reduce the risk of isolation [25], a study on the distribution of relief supplies after a disaster outbreak in isolated areas [6], and a study on efficient road or bridge restoration scheduling [26]. After reviewing methodological studies on the detection of isolated regions, Sheng et al. identified isolated communities in complex networks through spectral analysis based on Laplacian matrices [27,28]. Although this method is useful for identifying isolated areas in a complex network, it does not extract the disaster-related information necessary for isolated areas.

In addition, the population in isolated communities must be estimated more accurately to minimize human casualties in the event of a disaster. In general, demographic information is stored in a hierarchy of census geographic units, which differ from country to country [29,30,31]. To protect privacy, household demographic information is not provided directly to the public but through its smallest geographic unit, the size of which differs according to the country [32]. However, because this information does not provide demographic information for building units, it has limited use in disaster situations. For this reason, population estimation techniques are necessary to identify isolated communities in a disaster situation. In a related study, Qiu et al. obtained high-accuracy results by estimating the population from building heights obtained from LiDAR data and building areas obtained from RS images [33]. The population count was estimated from census data. Lwin and Murayama estimated the population by the ratio of the total floor area of each building to the census survey unit area [34]. Even if an isolated community is detected and the population in the isolated area is accurately estimated, previous technologies and efforts are useless without a disaster management system that can utilize this information. In addition, if there is no platform with which local residents and disaster managers can efficiently utilize estimated spatial information, such information is meaningless.

Web GIS has frequently been used for efficient disaster management in the GIS field [35,36]. A well-known application tool is OpenStreetMap (OSM), which provides free digital topographic maps and satellite images in a browser-based application [37]. In particular, Humanitarian OpenStreetMap Team (HOT) allows local communities and disaster management agencies to voluntarily map areas where disasters have occurred. OSM is a web map database built through volunteered geographic information (VGI), which is collected by way of crowdsourcing, a powerful way to produce the spatial data necessary for disaster management [8,38]. The advantage of web maps and OSM is that anyone, including developers, volunteers, disaster managers, and stakeholders, can easily participate in the web mapping environment needed for disaster relief [39,40]. The biggest advantage of OSM is that it can be used effectively for disaster management because it provides invaluable spatial information to individuals and rescue organizations in near real time [41]. Because of these advantages, various studies that use OSM or web maps have been conducted, including a study on quickly estimating damage from flooding [8], a study on creating a damage map using SNS data [40], and research on providing evacuation information on a web map built by volunteers [42].

Although various approaches to disaster management have been taken, there are still gaps in the literature. As noted in the Section 1, disaster management agencies need to focus on identifying and monitoring communities that are prone to isolation. First, although isolated communities are less urbanized and have smaller populations, the damage to human life and the natural environment that occurs should be approached from the same point of view as the disaster response methods used in large cities. Second, the population counts stored in census data at the smallest geographic level are insufficient for the microlevel analysis of isolated communities in a disaster situation. Thus, a method to accurately estimate the population living in isolated areas in advance or immediately after an actual disaster is needed. Furthermore, it is important to provide optimal evacuation routes to isolated local residents and to ensure that they are effectively guided to nearby shelters. Finally, it is necessary to develop an integrated disaster management system that can help residents of isolated communities by integrating the above three themes.

Consequently, although many researchers have focused on developing algorithms for disaster management or case studies on a type of disaster, few have focused on identifying isolated communities that are surrounded by mountains or countryside, particularly areas where disasters have often occurred. To fill this gap and answer the questions posed in the Section 1.1, this study proposes an automated analysis system that can be used for disaster response and recovery by extracting isolated risk areas in real time from a web-based GIS application. Specifically, this research first develops an algorithm to automatically detect communities at risk of isolation affected by disasters. In the following step, two analytical procedures are developed and included in the algorithm to identify buildings and populations within those communities and efficiently place at-risk residents in shelters. This web-based GIS platform enables residents or stakeholders in isolated communities to obtain useful disaster-related spatial information.

## 2. Materials and Methods

### 2.1. Materials

#### 2.1.1. Study Area

This study selected three rural communities surrounded by mountains where disasters have occurred in the past (Figure 1).

Figure 1a shows Nae-myeon, Hongcheon-gun in Gangwon-do, where a bridge was inundated and damaged due to heavy rain in 2018. Figure 1b shows Samsan 1 ri, Uljin-gun in Gyeongsangbuk-do, which was damaged by Typhoon Mitag in 2019. Figure 1c shows a portion of Imdong-myeon, Andong City, where areas were burned by wildfire in 2021. These areas have been damaged by unforeseen disasters in the past, which resulted in isolated communities and isolated residents.

#### 2.1.2. Data Collection

To ensure the reliability of data, we used data provided by the Korean government. All the data have high locational accuracy, are in GIS vector format, have been recently updated, and are stored in a geodatabase. Table 1 shows the dataset used in this research.

First, the centerline of the road was used to perform network analysis and investigate isolated areas and new routes for evacuation (Road in Table 1). These road data contain all types of roads, from local roads to highways. The attributes of the road include from and to nodes, distance, and travel times. Bridge information was also used to assume a disaster situation that causes isolation because of damaged bridges. The attribute table in the Bridge dataset includes the type, width, and length of the bridge. The two datasets were provided by the Korean National Geographic Information Institute (KNGII) [43], the national mapping agency of Korea. Building data, which were formatted as a polygon feature, were used to estimate the population that needs to be evacuated or helped in the isolated community. In terms of the population count, this study used census data, which were scaled at the smallest administrative boundary in the current hierarchy of census geographic units. This study used the census data officially updated in 2020 and provided by Statistics Korea [44]. Point features that indicate the locations of shelters were also used to estimate the number of residents in the area where the disaster occurred. The data were provided by the Ministry of the Interior and Safety, which is a branch of the government of South Korea [45].

All datasets were georeferenced to UTM-K with GRS 80 ellipsoid. The analysis outputs were reprojected to WGS84 for the web map developed in this study. For the base map, satellite images and topographic images available in OSM were used. All analytical procedures were implemented using the Python scripting language.

### 2.2. Methods

#### 2.2.1. Schematic Diagram for this Study

As introduced in the Section 1.1, this study is structured as follows: research background, related studies, data collection, methods, results, discussion, and conclusions. Figure 2 shows the workflow of this research.

In this study, three modules were developed. In the event of heavy rains, typhoons, or wildfires, it is assumed that bridge collapse, road flooding, forest fires, and farmland fires occur in rural communities. Figure 2a shows the first module, which automatically extracts isolated communities based on the working scenario, together with a list of data used in the working scenario. Figure 2b includes primary analytical procedures that automatically and rapidly estimate the number of buildings and the population in the isolated areas and even explore the closest shelters and provide evacuation routes to residents. In the second module, a Python script was used to conduct spatial data analysis. The last module is associated with a web-enabled GIS application that was designed for local residents and disaster management agencies. It was developed using JavaScript, which enables the creation of a web server. With the web-enabled GIS system (Figure 2c), the web-enabled GIS application can provide meaningful information to residents in need of help during disasters. Even citizens at risk during a disaster can access the system.

#### 2.2.2. Development of the Three Modules

First module: Extracting isolated communities affected by disaster

The first module, which is introduced in Figure 2a, includes six procedures to automatically extract and estimate the location of isolated regions. First, road data in a target area are input as a polyline that includes nodes and vertices (Figure 3a). Second, all roads in a network are separated into multiple subgraphs (Figure 3b). A subgraph is a subset of the nodes in a network and contains all the edges that connect these nodes [46]. The subgraphs are used later to extract isolated regions. To determine the location of the accident (Figure 3c), it is assumed that residents report the location of the damaged bridge, and the reported location is represented on the map input using the API in advance.

For the next step, we programmed a procedure to automatically search for subgraphs (Figure 3d) at the location of the accident and formed a convex hull that was later used to estimate the boundaries of the isolated regions. Figure 3e illustrates an estimate of an isolated area of 0.2 km^2^. In general, areas that tend to be isolated and surrounded by mountains are much smaller than urban areas. Thus, we used a base area of 0.64 km^2^ (Figure 3e) to determine the optimal size of the isolated area. The size was determined by surveying the largest area among the rural areas that were isolated in the countryside. The size of the area could be adjusted at any time considering the actual area of the community. As a result, Subgraph B was extracted as an isolated area (Figure 3f).


B.Second module: Analytical procedures for disaster response

The second module developed in this study includes four analytical procedures: estimating the number of buildings, estimating the population count, extracting evacuation routes, and searching for nearby shelters for people in the isolated areas. The first two steps estimate how many buildings and people are in the isolated area. This information is important because it is used by disaster managers to determine how many people a nearby shelter can accommodate.

Figure 4 shows the equation used to estimate the population in an isolated region.

To calculate the population in each building, this study used the ratio of building floor areas in the community to the census survey unit area. For example, after detecting an isolated area in a disaster-prone area, it is possible to accurately estimate the number of people living in the isolated area by knowing the number, type, and floor area of buildings in that area as well as the population size provided by census data. Generally, if the number of buildings in an isolated area is known, the number of people living in the buildings can be estimated from information provided to the National Statistical Office or local governments. However, the exact number of people living in each building is usually not provided directly to the public due to privacy concerns. As for the population data that are provided to the public, only the estimated population data for the smallest spatial unit (i.e., block level) in the census data are provided. Because the purpose of this study is to develop a disaster management system that can be used by both the public and disaster management agencies, the number of people was accurately estimated by considering the proportions of the number of people recorded at the smallest spatial scale of census data, the type of buildings, and the floor area of all buildings in the isolated area [33].

As shown in Figure 4, it is assumed that there are three buildings in a rural area, and two of them are detected as buildings in the isolated area (extracted from the algorithm introduced in Figure 3). The sizes of the three buildings are 50, 30, and 20 m^2^, respectively. The sizes of the two buildings in the isolated areas are 50 and 30 m^2^, and the rural area has a total population of 30 people. If the total population is divided by the ratio of the building area, the number of people living in each building is 15, 9, and 6. Applying this calculation method, the affected population in the isolated area can be estimated as 24.

Figure 5 below shows the diagrams for the Python script programs.

Figure 5a shows a diagram that represents how to detect the isolated areas that are addressed in the first module. Figure 5b illustrates the process for identifying buildings in an isolated community and estimating the population in the community. After the number of buildings and the population in the isolated community are estimated, it is important for the population to immediately evacuate the areas and rapidly move to a nearby shelter.

Table 2 below shows pseudocode that finds an evacuation route from the isolated areas to a nearby shelter.

The algorithm considers the preferred distance and time people are likely to walk. The system developed in this research first searches for a nearby shelter. This search is performed by the Dijkstra algorithm, which is an algorithm for finding the shortest path between nodes in a subgraph and is a type of A* algorithm [47]. This algorithm enables us to find the shortest path between nodes and prioritizes which paths to explore. In the next step, we consider the number of people each shelter can accommodate. The process of allocating people to shelters considering the number of people in the isolated area is included in the development stage. For example, the process is designed to provide information about the nearest shelter to people in the quarantined area first and to provide information on the next shelter if the number of people in the isolated area exceeds the number of people that the shelter can accommodate. Evacuation routes are created from each of the buildings to increase visibility and usability.

C.Third module: Developing a web-enabled GIS application

For the last module, we developed a web-enabled GIS application that can be used by the public and by stakeholders who want to immediately acquire spatial information associated with a disaster. The web-based GIS application allows residents to quickly search for evacuation routes and disaster managers to instantly estimate the number of buildings and people in isolated communities.

Table 3 below shows the pseudocode of the web mapping application we developed in this study.

The third module is programmed in JavaScript to support residents and disaster management agencies. The web environment contains all the analytical procedures developed in the first and second modules.

## 3. Results

### 3.1. Real Application with Three Communities where Disasters Have Occurred in the Past

As mentioned in Section 2.1, three areas where disasters have occurred in the past were selected (heavy rain in 2018, a typhoon in 2019, and a wildfire in 2021). The disasters in the three areas happened unexpectedly and produced isolated areas where residents were quarantined.

Figure 6 shows Gwangwon-ri of Nae-myeon, Hongcheon-gun in Gangwon-do, which was damaged by heavy rain.

The yellow line in Figure 6a depicts where the bridge was inundated and damaged due to heavy rain in 2018. The red line depicts isolated areas and roads. As introduced in the Section 2.2, after an accident point is detected or reported, subgraphs in the network are automatically extracted and form a boundary that represents isolated areas. Figure 6b illustrates the buildings and the populations in the isolated areas. The blue line in Figure 6c depicts the evacuation routes from all buildings in the isolated areas. Figure 6d shows the evacuation route and the nearest shelter that evacuees of each building can reach. Moreover, the estimated number of evacuees is allocated to the nearest shelter. The nearby shelter can accommodate up to 225 and is large enough to accommodate the 36 refugees living in isolated areas.

Table 4 shows the number of possible routes and the maximum (Max.), minimum (Min.), and average (Ave.) distance and travel time in the isolated region.

Twenty-nine buildings were identified in this isolated area (Figure 6a), and 29 possible evacuation routes were provided from the buildings. The longest evacuation route was 7.63 km, the shortest was 3.39 km, and the average was 4.18 km. The maximum travel time was 7.63 min, the minimum was 3.39 min, and the average was 4.18 min. To compute the travel time by vehicle, a speed limit of 60 km/h was applied for all roads in rural areas. Thus, the same speed limit was applied in all study areas.

Figure 7 shows the second study area where a disaster has occurred in the past.

The maps in Figure 7 illustrate Samsan 1 ri, Uljin-gun in Gyeongsangbuk-dom, which was damaged by Typhoon Mitag in 2019. Typhoon Mitag was a moderately strong tropical cyclone that severely affected South Korea from 1 October to 6 October 2019. It caused 12 deaths, 3 missing persons, 11 injured persons, and 910 refugees and significant property damage. During this period, Uljin-gun had a high cumulative precipitation of 582.8 mm, and as a result, four people died, 124 roads were damaged, and 98 river bridges were damaged [48].

As shown in Figure 7a, a bridge was damaged, and isolated areas (red line) were identified by the algorithm developed in this research. Seven buildings and nine refugees were detected as being isolated (Figure 7b). As in the previous steps, the next step was to identify the nearest shelter in the surrounding area. Accordingly, the populations at risk of isolation were efficiently assigned to the evacuation routes (Figure 7c), and the refugees were assigned to the nearest of the four shelters considering the total number of refugees (Figure 7d).

Table 5 below illustrates the length of the evacuation route and the travel time by vehicle from the isolated area of Samsan-ri to the nearest shelter.

Seven buildings were identified as being isolated, and seven possible evacuation routes were also provided. The longest evacuation route was estimated to be 3.74 km, the shortest was 3.59 km, and the average was 3.69 km. The maximum travel time was 3.74 min, the minimum was 3.39 min, and the average was 3.69 min.

Figure 8 shows where a wildfire occurred in 2021.

The yellow boundary in Figure 8a indicates mountains affected by a wildfire. The wildfire caused approximately KRW 8 billion (won) in property damage, and approximately 307 hectares of mountainous area were burned [49]. The isolated areas were detected by the boundary of the wildfire (inside the yellow boundary in Figure 8a) as reported by firefighters. From the boundary, isolated areas (subgraphs) were classified by the algorithm developed in this study.

As shown in Table 6, 209 buildings and 492 refugees were classified as isolated (Figure 8b). Ten shelters were suggested to accommodate all the refugees. The evacuees were assigned to the 10 shelters sequentially based on shelter distance and size. As shown in Table 6, the refugees were assigned to six shelters accordingly: Shelter ID 1, 3, 5, 6, 7, and 8.

Table 7 illustrates the length of the evacuation route and the travel time by vehicle from the isolated area of Jeungpyeong-ri to the nearest shelter.

According to the computational procedure developed in this study, 209 possible evacuation routes were provided. The longest evacuation route was 6.39 km, the shortest was 1.34 km, and the average was 5.03 km. The maximum travel time by vehicle was 6.39 min, the minimum was 1.34 min, and the average was 5.03 min.

Figure 9 compares in graphic format the time taken for all refugees in the three isolated communities to move to a shelter.

In the isolated area in Gwangwon-ri, the first evacuation was completed in 3.39 min, and all refugees were evacuated within 7.63 min (Figure 9a). This area was found to require the longest travel time due to the nature of the area. Of the total 29 evacuation routes, 20 of them provided a shelter reachable within 4 min, indicating that most of the evacuations were completed in 3–4 min. In the second study area (Figure 9b), the first evacuation was completed in 3.59 min, and all evacuations were completed within 15 s after that. In the last study area (Figure 9c) the first evacuation was completed in 1.34 min, and the evacuation speed was much faster than in the other two areas. All refugees were able to reach a shelter within around 5.03 min.

To summarize, the results obtained through the three simulations, for communities prone to isolation, it was found that the evacuation time and arrival time differed depending on the characteristics of the road structure and the number of shelters available. The results of this analysis indicate that when an actual disaster occurs, a differentiated disaster support policy is necessary in consideration of the physical or environmental characteristics of the region. If disaster management agencies are faced with a disaster without prior training, they will be unable to carry out safe and effective disaster recovery operations, which will ultimately result in huge loss of life and property. In particular, in the case of an area with a high probability of isolation surrounded by mountains, there are more physical restrictions on recovery activities than in an urban area, so a disaster recovery strategy specialized for rural areas is essential. In conclusion, the results obtained through this study are expected to be important disaster-related spatial information that can enable effective responses to disaster situations that may occur in the future. The next section introduces the development of a web-based disaster management system that disaster managers and local residents can use together.

### 3.2. Web-Enabled GIS Application

As introduced in the previous section, in order to effectively respond to disaster situations, it is important to utilize a platform that enables local residents as well as disaster managers to acquire disaster-related information in real time and make decisions directly. In this way, it is possible to overcome disaster situations more safely and efficiently by using an environment in which disaster managers and local residents can directly participate and make their own decisions through a bottom–up rather than top–down approach. Therefore, this web-based GIS platform enables residents in isolated areas to obtain disaster-related information for their area. Consequently, the platform enhances public engagement in disaster management.

Figure 9 shows the web-enabled GIS application developed in this study.

As described in previous sections, all analytical procedures were integrated into the web-based GIS application. The web browser was developed using JavaScript, and all analytical processes were implemented using Python. This application includes two major analytical functions to find the location of a disaster: damage estimation and evacuation route creation. The web map uses OSM, which is a freely available database. The OSM-based web map includes roads, the locations of shelters, legal boundaries, buildings in the study area, and base satellite images.

Figure 10a shows an example of the estimation of buildings and populations in isolated areas. The DAMAGE ESTIMATION button is first clicked, and then the RUN button is clicked. Isolated areas (subgraphs) are selected. The process is implemented through a Command Prompt (CMD) window. After the process is complete, the OPEN MAP button is clicked, and the isolated areas are displayed on an OSM-based web map. In the next step, buildings within the boundary of the shapefile are selected, and the population in isolated risk areas is automatically counted. In the right corner, an output layer is added to the legend, and this layer can be freely toggled on and off. After the first analysis phase is complete, the end-user knows how many buildings and people are in the isolated area. The next analytical function is evacuation route creation (Figure 10b). This process includes working steps that are similar to the previous steps. When an end-user selects the shapefile of isolated areas in a CMD window, the EVACUATION ROUTES button is activated. With the OPEN MAP button, evacuation routes and the locations of shelters are estimated and displayed on the OSM-based web map. Because the OSM-based web map is designed to complete the analysis by specifying the shapefile of the isolated areas, it is quite easy for individuals and stakeholders to use. In addition, because the analysis result map can be saved as an HTML file, it can be used sustainably.

In summary, the web-enabled GIS environment implemented through the results of this study allows local residents and disaster managers to easily access this environment to check disaster-related information and to directly find the optimal route or shelter using spatial analysis tools. Through this environment, local residents will be able to make their own decisions when an actual disaster situation occurs and thus respond quickly to disasters.

## 4. Discussion

Human casualties and damage to the natural environment caused by disasters occur every year, requiring continued efforts to repair damages from unexpected disaster events [50]. Because unpredictable damage is caused by extreme weather, such as heat waves and floods, a sustainable disaster management strategy is needed to prepare for rapid changes in the environment [51,52,53]. As introduced in the Section 1.2, to stably manage isolated hazardous areas, it is necessary to identify isolated communities in advance and to predict the number of buildings and the population in the isolated communities. In addition, it is essential to develop a sustainable disaster management system for isolated areas that can provide optimal evacuation routes for refugees and efficiently distribute them in a timely way. This means that research on isolated communities in actual disaster situations should be approached from the same point of view in all areas, regardless of the size of the city or the degree of urbanization. In other words, sustained attention and effort are needed for areas surrounded by mountains or areas that are likely to be isolated. Although countless studies have used RS and GIS to effectively respond to disasters, few disaster-related studies have focused on communities that are prone to isolation. As noted earlier, this study was conducted to answer the four questions raised in the Introduction: What are the essential geospatial information technologies for communities prone to isolation from disasters? Moreover, what analytical procedures are required for these areas? Next, what is the best way to deliver disaster-related information to citizens and disaster managers in real time? Finally, can scenario-based analytics help isolated communities at risk of disasters in real situations? To fill the gap we identified from previous studies, we decided to develop an automated analysis tool for isolated communities that can be used at the same time by residents and disaster management agencies. In particular, the analysis was performed assuming that an actual disaster occurred based on the scenario. Next, an analysis tool was developed that could select an isolated area using an automated technique, identify the number of buildings in the isolated area, and estimate the number of residents. Moreover, an analysis process was developed to enable the efficient distribution of expected evacuation routes and shelters for refugees. All these analytic processes were developed in Python and integrated into a web-based GIS environment that residents and disaster management organizations can easily use.

Although we developed a well-suited disaster management system for isolated communities, there are still some points that need improvement. For example, the method used for population estimation that considers the ratio of the building floor area and the population is more effective than the approach that uses existing census data. However, a more accurate population estimate would be possible if one could consider the number of floors in a building or directly utilize the demographic data stored on a particular building. As is commonly known, demographic data for building units is not publicly available because of privacy concerns. In addition, to better utilize the disaster management system in isolated areas through the web map, it is necessary to build a web map environment that can be supported in a mobile environment, and more diverse analysis tools (gadgets) need to be developed. In addition, integrating disaster-related statistical information for a region with the disaster management system developed in this study would yield information that would be extremely helpful in establishing a sustainable management system.

In summary, this study makes three contributions. First, few studies have selected only isolated communities, and no studies have investigated a local community that was surrounded by mountains and that had a high risk of isolation. Therefore, this study offers an effective analysis tool for detecting isolated regions across the country in advance. Next, using the results of the study based on the scenarios suggested in this study enabled the expected number of isolated people and the number of isolated households to be determined to prepare for future disasters. It is expected that these data could be used as basic data when an actual disaster occurs. Finally, residents and disaster management agencies can use an integrated analysis tool at the same time through a web-enabled GIS application. From a resident’s point of view, they can find the optimal evacuation route and know which shelter they should go to, which helps in decision-making. It will enable disaster management agencies to assist local residents and restore isolated communities through the scenario-driven experience. When a disaster occurs in a mountainous region that lacks communication and transportation infrastructure, we expect that this tool will accurately detect isolated areas and be used to support disaster response decision-making regarding, for example, rescue activities and supply of materials.

## 5. Conclusions

This study aimed to develop an automated analysis system that could be used for disaster response and recovery by extracting isolated risk areas in real time from a web-based GIS application. A variety of disaster scenarios were defined for mountainous and rural areas where disaster response is difficult due to geographical limitations. With the contributions of this study, we are confident that automated analytics systems will benefit both people living in isolated hazardous areas and disaster managers or relevant stakeholders responsible for disaster response and recovery. Although we are convinced of the scientific value of the results obtained through this study, we also identified areas for improvement. For example, as detailed in the Section 4, there is a need to develop mobile-based applications that both residents and disaster managers can easily use. In addition, it is necessary to develop a more accurate and precise population estimation method. Therefore, in future studies, we plan to develop a mobile-based disaster management support system and a more precise population estimation method for isolated communities across the country.

## Figures and Tables

**Figure 1 ijerph-19-13995-f001:**
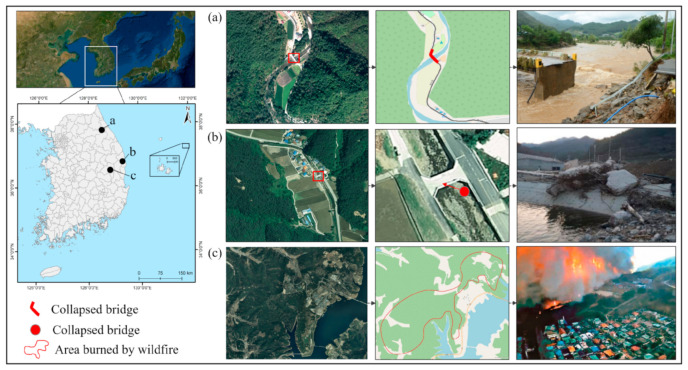
Study areas in South Korea: (**a**) Nae-myeon, Hongcheon-gun; (**b**) Samsan 1 ri, Uljin-gun; and (**c**) a part of Imdong-myeon, Andong City (photo courtesy of Korea Forest Service, 2021).

**Figure 2 ijerph-19-13995-f002:**
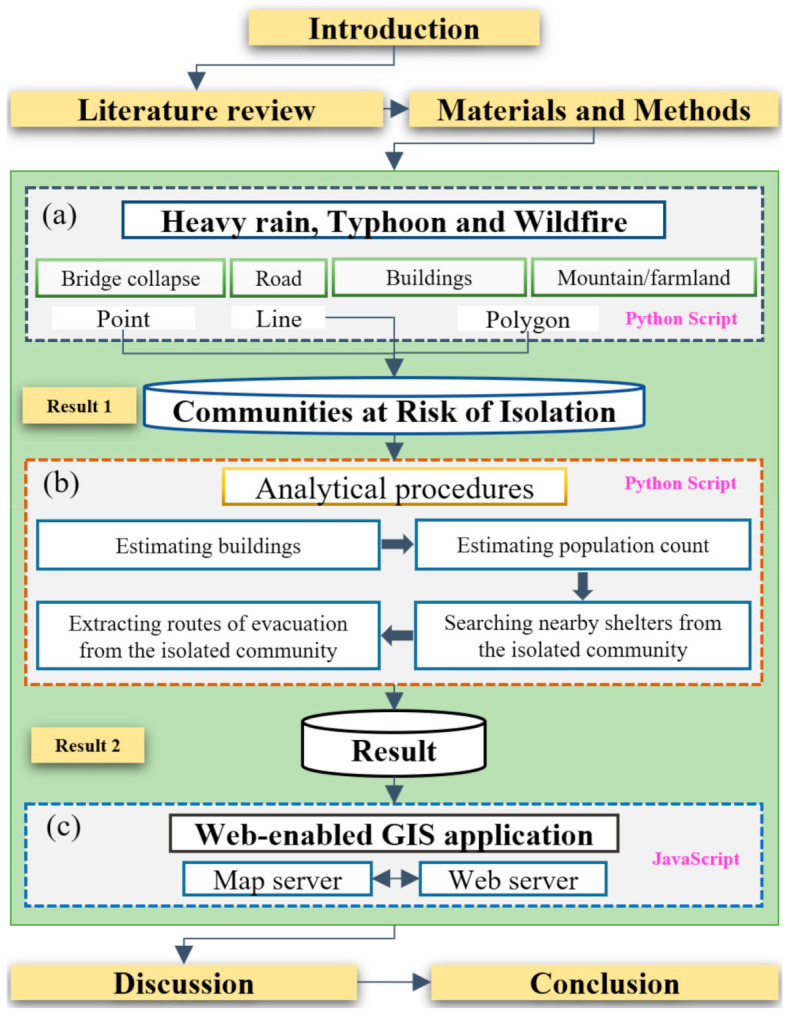
Workflow of this research: (**a**) first module: Data collection and working scenarios, (**b**) second module: Analytical procedures, (**c**) third module: Deployment of web-enabled GIS application.

**Figure 3 ijerph-19-13995-f003:**
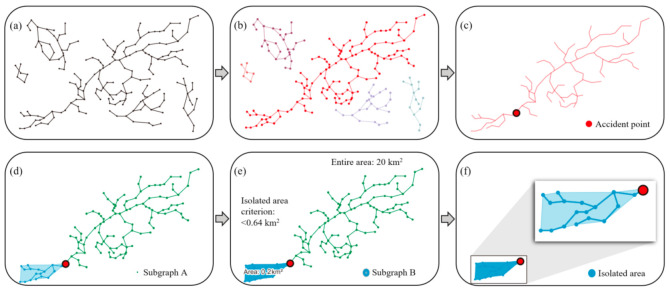
Extracting isolated communities: (**a**) roads; (**b**) multiple subgraphs; (**c**) accident point; (**d**) a subgraph selected; (**e**) a subgraph as polygon; (**f**) an isolated area.

**Figure 4 ijerph-19-13995-f004:**
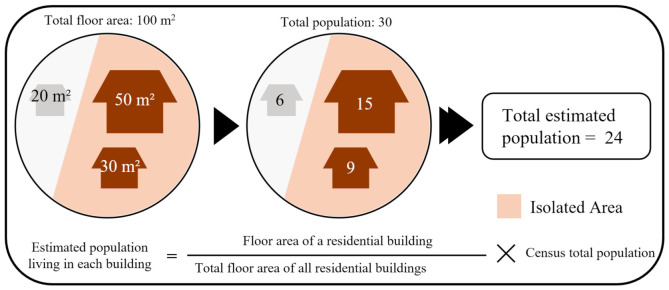
Estimating the population in an isolated area.

**Figure 5 ijerph-19-13995-f005:**
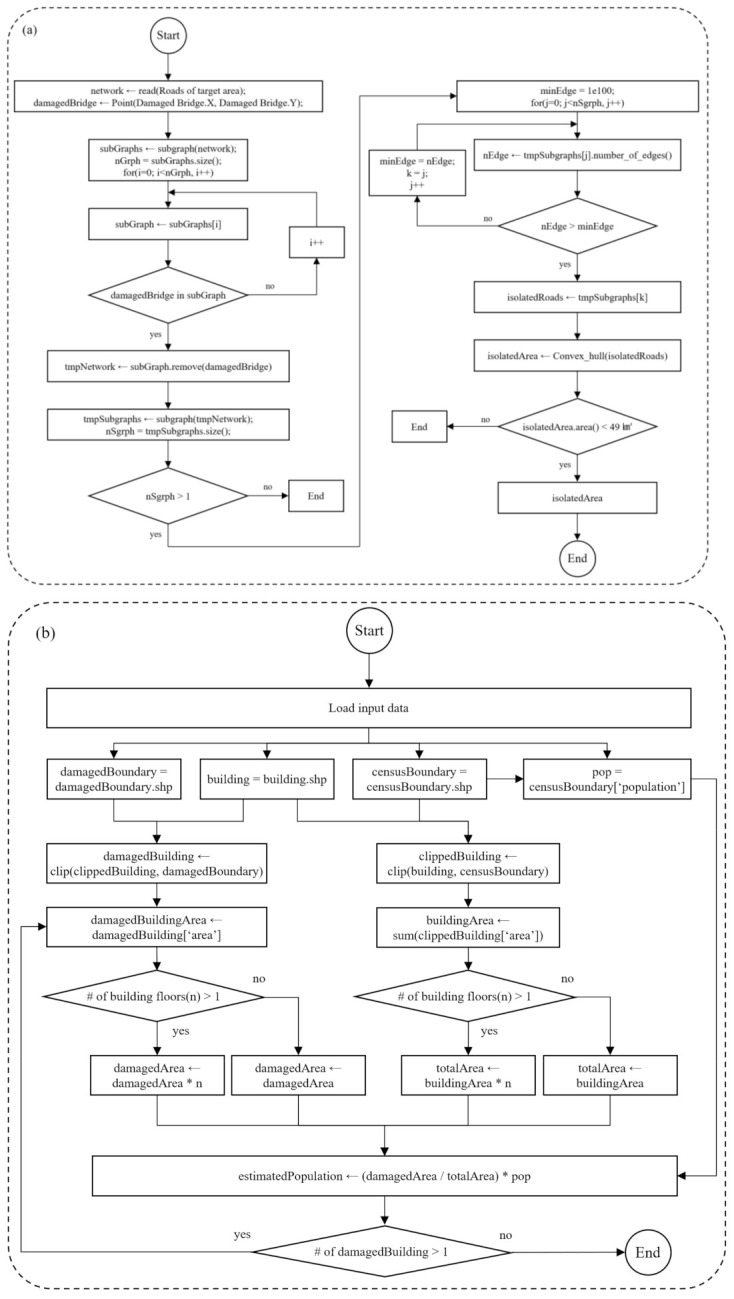
Diagrams showing the two Python scripts programmed to detect (**a**) the boundary of the isolated area and (**b**) the buildings and population in the isolated area.

**Figure 6 ijerph-19-13995-f006:**
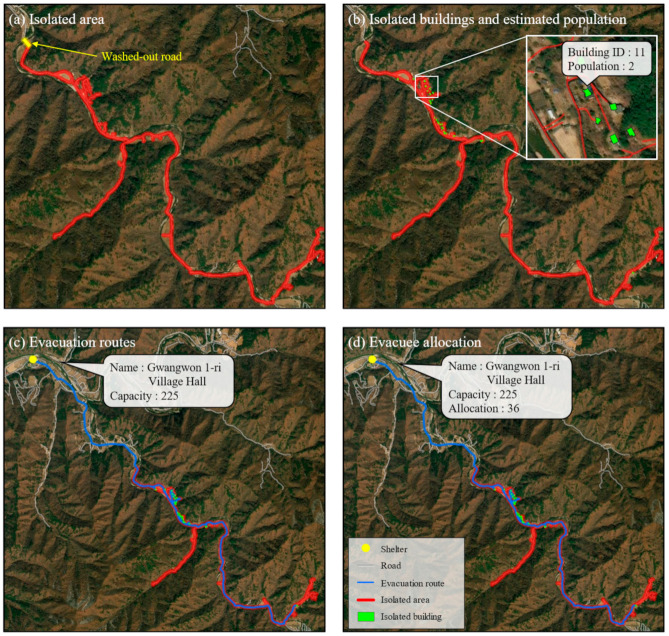
Mapping result for an area where a disaster has occurred in the past: Gwangwon-ri of Nae-myeon, Hongcheon-gun in Gangwon-do: (**a**) Isolated area; (**b**) buildings and population in the isolated area; (**c**) possible evacuation routes; (**d**) the nearest shelter for evacuees.

**Figure 7 ijerph-19-13995-f007:**
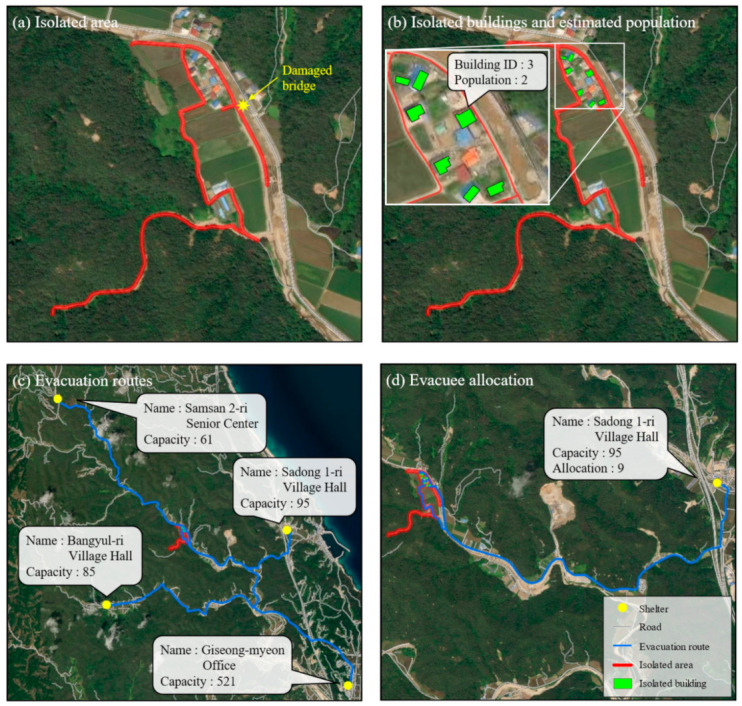
Mapping results for an area where a disaster has occurred in the past: Samsan 1 ri, Uljin-gun in Gyeongsangbuk-do: (**a**) Isolated area; (**b**) buildings and population in the isolated area; (**c**) possible evacuation routes; (**d**) the nearest shelter for evacuees.

**Figure 8 ijerph-19-13995-f008:**
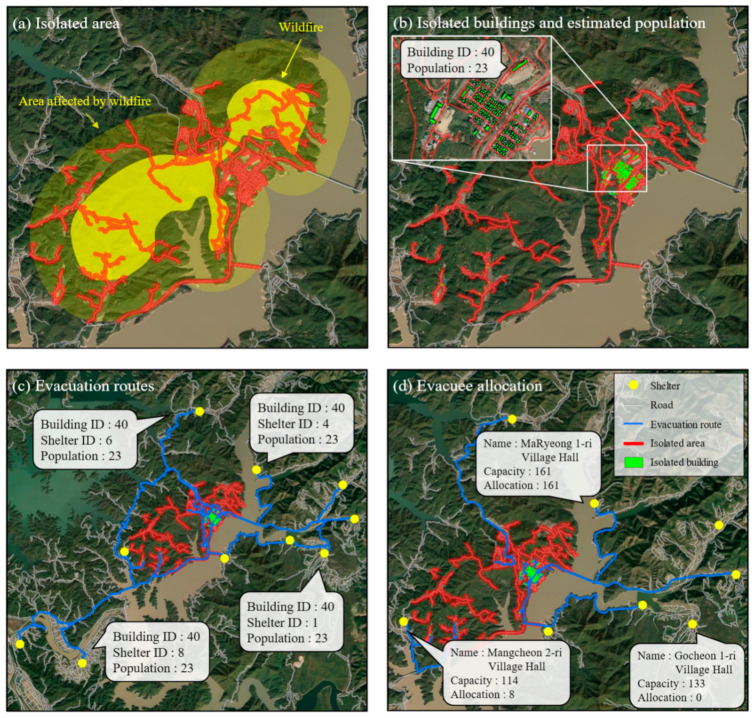
Mapping results for a location where a disaster occurred in the past: Imdong-myeon, Andong City: (**a**) Isolated area; (**b**) buildings and population in the isolated area; (**c**) possible evacuation routes; (**d**) the nearest shelter for evacuees.

**Figure 9 ijerph-19-13995-f009:**
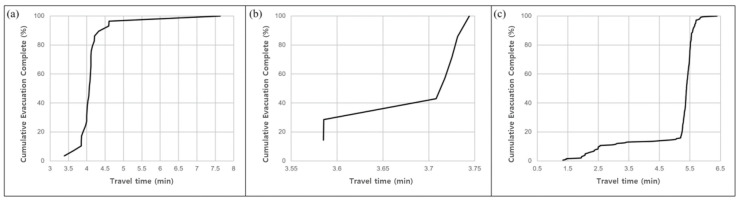
Evacuation rate over time for the three areas affected by (**a**) heavy rain (Gwangwon-ri), **(b**) typhoon (Samsan 1 ri), and (**c**) wildfire (Imdong-myeon).

**Figure 10 ijerph-19-13995-f010:**
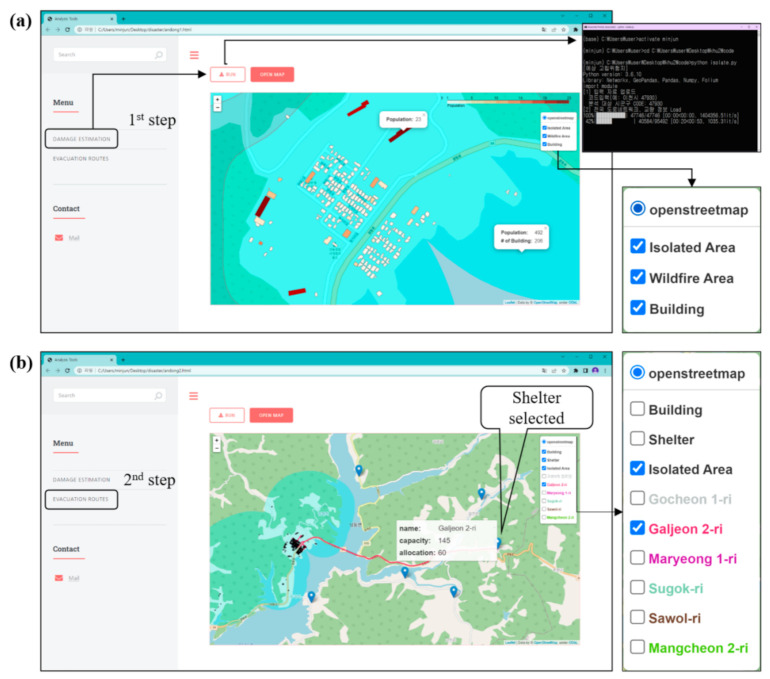
Web-enabled GIS application: (**a**) first function: damage estimation; (**b**) second function: evacuation route creation.

**Table 1 ijerph-19-13995-t001:** Data collection and source.

Name	Feature Types	Source
Road	Polyline	National Geographic Information Institute
Bridge		
Building	Polygon	Ministry of Land, Infrastructure, and Transport
Shelter	Point	Ministry of the Interior and Safety
Population	Text	Statistics Korea

**Table 2 ijerph-19-13995-t002:** Pseudocode for the derivation of evacuation routes.

**#package networkx, pandas, geopandas, json, numpy** **#Input data** Input: building ← Buildings, shelter ← Shelters, network ← Road networks **#Output data** Output: routes ← Evacuation route each building shelter_pop ← The number of people assigned to each shelter. **#Calculate network geometry** FOR EACH edge IN network DO edge.length = distance(edge[a], edge[b]) ← Distance between road nodes END FOR **#Shortest route extraction and allocation** FOR EACH building IN buildings DO FOR EACH shelter IN shelters DO route ← Dijkstra algorithm route append route to routes append pop to shelter_pop END FOR END FOR

**Table 3 ijerph-19-13995-t003:** Pseudocode of the web map settings.

**#Convert GeoJSON and load** GDF.to_file(‘FileName.json’, driver = ‘GeoJSON’) GeoJSON = json.load(open(‘FileName.json’)) **#Basemap settings** Center = list(reversed(list(GDF[‘geometry’].centroid [0].coords [0]))) baseMap = folium.Map(location = center, tiles = None, overlay = False, zoom_start = n) **#Create a layer group and add layer** group = folium.FeatureGroup(name = ‘uljin’, overlay = False).add_to(baseMap) building = plugins.FeatureGroupSubGroup(group, name = ‘building Layer’, show =False).add_to(baseMap) population = plugins.FeatureGroupSubGroup(group, name = ‘evacuation Layer’, show = False).add_to(baseMap) **#Set and apply object style** style1 = {‘color’:’black’, ‘fillColor’:’#B22222, ‘weight’:0.5, ‘fillOpacity’:1.0} style2 = {‘color’:’red’, ‘weight’:1, ‘opacity’:0.5} area = folium.GeoJson(building, style_function = style1, show = False).add_to(baseMap) **#Create a marker** marker = folium.Marker(location = [lat, lon], popup = ‘marker’, icon = g.Icon(color = “red”)).add_to(baseMap) **#Pop-up window settings** popup = (‘<div style = “font-size: 15pt” > ‘ + “population:{pop}” + ’</div > ‘).format(pop = feature[‘properties’][‘POP’])

**Table 4 ijerph-19-13995-t004:** Length of the evacuation route and the travel time by vehicle from the isolated area of Gwangwon-ri to the nearest shelter.

Possible Routes	Distance (km)			Estimated Travel Time (Minutes)		
	Max.	Min.	Ave.	Max.	Min.	Ave.
29	7.63	3.39	4.18	7.63	3.39	4.18

**Table 5 ijerph-19-13995-t005:** Length of the evacuation route and the travel time by vehicle from the isolated area of Samsan-ri to the nearest shelter.

Possible Routes	Distance (km)			Estimated Travel Time (Minutes)		
	Max.	Min.	Ave.	Max.	Min.	Ave.
7	3.74	3.59	3.69	3.74	3.59	3.69

**Table 6 ijerph-19-13995-t006:** Capacity and allocation of shelters by evacuation route in the isolated areas of Jeungpyeong-ri.

Shelter ID	Shelter Name	Capacity	Allotted Evacuees	Allotted Buildings
1	Gocheon 1-ri Senior Center	163	163	80
2	Gocheon 1-ri Village Hall	133	0	0
3	Galjeon 2-ri Village Hall	145	60	19
4	Galjeon 1-ri Village Hall	126	0	0
5	Maryeong 1-ri Village Hall	161	161	81
6	Sugok-ri Village Hall	98	98	20
7	Sawol-ri Village Hall	106	2	2
8	Mangcheon 2-ri Village Hall	114	8	7
9	Imha 1-ri Office	119	0	0
10	Chuwol Village Hall	81	0	0
#	Total		492	209

**Table 7 ijerph-19-13995-t007:** Length of the evacuation route and the travel time by vehicle from the isolated area of Jeungpyeong-ri to the nearest shelter.

Possible Routes	Distance (km)			Estimated Travel Time (Minutes)		
	Max.	Min.	Ave.	Max.	Min.	Ave.
209	6.39	1.34	5.03	6.39	1.34	5.03

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
