# Peer review of "Developing an Automated Analytical Process for Disaster Response and Recovery in Communities Prone to Isolation"

_ijerph, 2022, doi:10.3390/ijerph192113995_

Round 1

Reviewer 1 Report

Major changes

1. The abstract should be re written. The abstract should state briefly the purpose of the research, the principal results and major
conclusions. The current version is too generic.
2. The Introduction should be rewritten. Basically, the Introduction should  clearly state the research questions and targets
first. Then answer several questions: Why is the topic important (or why do you study on it)? What are research objectives?
What has been studied? What are your contributions?
3. There are a lot of similar articles published in reputed journals with similar research ideas. I do not find any distinct difference and novelty in this manuscript. Try to highlight your novelty.
4. Literature review is should be done systematically. Authors should be describing on what basis literature review
was done? Articles from which time frame, journals etc were shortlisted for this study? Also, currently, the literature
review section is not building the gap for your research.
5. Many challenges are not listed out and this needs attention. 

6. Provide a research flowchart for the readers to easily follow the manuscript.
7. The manuscript has too much colloquialism and with the lack of a proper order in the content,
it is very difficult to follow. Authors have to rethink a different flow. I may suggest them to
go for Introduction, Literature review, Research gap analysis, Research objectives and framework,
Methodology, Result and Discussion, Conclusion.
8. It is important to discuss in detail about the data collection techniques used 
9. The authors have failed to clearly explain the implications of the results. Improve the findings section with critically
addressing the applicability of your findings
10. The discussion on results is poorly presented. The execution of the proposed methodology is appreciable while the
discussion of the obtained results must be well improved highlighting the insights of the research findings and with
support from earlier literature. I find many unsupported statements in this section.
11. Please revise your conclusion part into more details. Basically, you should enhance your contributions, limitations,
underscore the scientific value added of your paper, and/or the applicability of your findings/results and future study in this session 

Author Response

Response to Reviewers

Dear Reviewer 1,

We would like to express our appreciation in terms of your direct and thoughtful comment. After reading the review comments, we could build up more strong academic foundation related to our research questions. In particular, we have carefully reviewed the paper again to ensure that we comply with the journal’s preference in subjective circumstances. Again, we have crosschecked all references, figures, and minor typo, grammar errors in the body of the text. All changes are in Yellow.

Thanks for your valuable and constructive comments to this paper.

The abstract should be re written. The abstract should state briefly the purpose of the research, the principal results and major conclusions. The current version is too generic.

  • As recommended, the abstract is concisely revised to avoid a generic explanation. `

The Introduction should be rewritten. Basically, the Introduction should clearly state the research questions and targets first. Then answer several questions: Why is the topic important (or why do you study on it)? What are research objectives? What has been studied? What are your contributions?

  • We did our best to revise it and has changed some of area.

Provide a research flowchart for the readers to easily follow the manuscript.

 ð  As suggested, Figure 2 has been modified to make it easier for the reader to follow the manuscript. 

The manuscript has too much colloquialism and with the lack of a proper order in the content, it is very difficult to follow. Authors have to rethink a different flow. I may suggest them to go for Introduction, Literature review, Research gap analysis, Research objectives and framework, Methodology, Result and Discussion, Conclusion.

  • Another reviewer also stated the order of the manuscript. The reviewer’s suggestion is a bit different from your order. So, we decided to follow the Journal’s suggestion. It has been revised as follows. Introduction, Materials and Methods, Results, Discussion, and Conclusions. To make this paper readable, we have revised several portions in the Introduction, Results, Discussion, and even the conclusion.
  • Moreover, to avoid colloquialisms, all sentences were reviewed by An experienced editor whose first language is English and who specializes in editing papers and has carefully reviewed the revised manuscript.

It is important to discuss in detail about the data collection techniques used

  • As suggested, we have updated the detailed information for data collection. To ensure the reliability of GIS data, we used the data created by the South Korean government.

  1. The authors have failed to clearly explain the implications of the results. Improve the findings section with critically addressing the applicability of your findings
  • As suggested, we have clearly explained the implications of the results at the end of Sections 3.1 and 3.2.

  1. The discussion on results is poorly presented. The execution of the proposed methodology is appreciable while the discussion of the obtained results must be well improved highlighting the insights of the research findings and with support from earlier literature. I find many unsupported statements in this section.
  • The discussion has been revised to improve readability. Furthermore, we did our best to highlight research questions and findings.

  1. Please revise your conclusion part into more details. Basically, you should enhance your contributions, limitations, underscore the scientific value added of your paper, and/or the applicability of your findings/results and future study in this session
  • As suggested, the conclusion is rewritten to clarify the contribution and limitations of this study in detail and to introduce future research plans.

We have also updated multiple portions recommended by another reviewer. We are confident this paper has been more academically improved. Thanks for your valuable and constructive comments on this paper.

Please stay safe and healthy, we look forward to hearing from you. 

Warm regards, 

Reviewer 2 Report

Dear authors,

The presented article is interesting and offers useful solutions for remote populations suffering from devastating disasters near their homes. However, I offer some suggestions for improvement:

- The abstract should indicate the study populations, as detailed in section 2. Materials.

- The keywords are in line with the topic of the study.

- In the introduction, the numbers relating to the research consulted should be unified. The numbers should be [10-13], instead of [10, 11, 12, 13] (line 91). Carefully check all the numbers in the document.

- On line 104, enter Sheng et al. [26]. Do not put the year. The same applies to other examples [32, 33, ...], check the whole document.

- Line 134 [36, 8]. The number 8 goes before 36 [8, 36]. Change it.

- Although the objectives of the study are outlined throughout the article, it is necessary to detail them one by one at the end of line 163. This makes it easier for the reader to understand the purpose and objectives of this work.

- In general, the "Introduction" section is very well-founded and sets out the subject of the study correctly. Sufficient previous studies are provided to support each of the ideas proposed throughout the text. Most of the previous studies are current and belong to the last 5 years.

- Section 2. Materials. I believe that this section should be linked to section 3. The appropriate title would be "Method and Materials", as it is usually seen in current research articles. I propose to merge both sections and renumber them.

- The figures are very illustrative and fit perfectly with the description of the text.

- Section 4. Results. The tables are in accordance with MDPI editorial standards. The results obtained are well written and free of errors.  The interpretation of the results is presented to the reader in a clear and orderly manner. 

- Section 5. Discussion. Although this section is extensive, I feel that it does not really discuss the results obtained. There is a very elaborate summary of the process followed and the results, but I do not feel that the results shown in the previous section are discussed one by one. Nor are any links established with the previous studies cited in the "Introduction" section.

- Section 6. Conclusions. The conclusions are brief and only present an elaborate summary of the work done. There is no in-depth reflection on natural disasters and on the usefulness of the automated analysis system. I believe that this section could be improved. 

- References section. In general, all references are in line with MDPI standards, but in the case of references to journals, the year should be put in "bold". For example, on line 520, 2019 would be in bold. Correct all references. 

The article has a broadly correct structure. It provides the reader with an automated analysis system that will undoubtedly be useful to the research community and, of course, to communities affected by natural disasters. I recommend that the suggested improvements be made.

Author Response

Response to Reviewers

Dear Reviewer 2,

We would like to express our appreciation in terms of your direct and thoughtful comment. After reading the review comments, we could build up more strong academic foundation related to our research questions. In particular, we have carefully reviewed the paper again to ensure that we comply with the journal’s preference in subjective circumstances. Again, we have crosschecked all references, figures, and minor typo, grammar errors in the body of the text. All changes are in Yellow.

Thanks for your valuable and constructive comments to this paper.

- The abstract should indicate the study populations, as detailed in section 2. Materials.

->  The abstract is concisely revised to avoid a generic explanation. `

- The keywords are in line with the topic of the study.

->  Thanks for the positive comment.  

- In the introduction, the numbers relating to the research consulted should be unified. The numbers should be [10-13], instead of [10, 11, 12, 13] (line 91). Carefully check all the numbers in the document.

->  All numbers are corrected as suggested. 

- On line 104, enter Sheng et al. [26]. Do not put the year. The same applies to other examples [32, 33, ...], check the whole document.

-> Those years are removed.  

- Line 134 [36, 8]. The number 8 goes before 36 [8, 36]. Change it.

->  Changed and reviewed through the paper.  

- Although the objectives of the study are outlined throughout the article, it is necessary to detail them one by one at the end of line 163. This makes it easier for the reader to understand the purpose and objectives of this work.

-> As suggested, the objectives of this study are introduced at the end of the line.

- In general, the "Introduction" section is very well-founded and sets out the subject of the study correctly. Sufficient previous studies are provided to support each of the ideas proposed throughout the text. Most of the previous studies are current and belong to the last 5 years.

-> Thanks for the positive comment.  

- Section 2. Materials. I believe that this section should be linked to section 3. The appropriate title would be "Method and Materials", as it is usually seen in current research articles. I propose to merge both sections and renumber them.

-> Thanks for the suggestion. As suggested, the two sections were combined into one section called "Materials and Methods".

- The figures are very illustrative and fit perfectly with the description of the text.

->  Thanks for the positive comment.  

- Section 4. Results. The tables are in accordance with MDPI editorial standards. The results obtained are well written and free of errors.  The interpretation of the results is presented to the reader in a clear and orderly manner. 

-> Thanks for the positive comment.  

- Section 5. Discussion. Although this section is extensive, I feel that it does not really discuss the results obtained. There is a very elaborate summary of the process followed and the results, but I do not feel that the results shown in the previous section are discussed one by one. Nor are any links established with the previous studies cited in the "Introduction" section.

-> The discussion has been revised to improve readability. Furthermore, we did our best to highlight research questions and findings.

- Section 6. Conclusions. The conclusions are brief and only present an elaborate summary of the work done. There is no in-depth reflection on natural disasters and on the usefulness of the automated analysis system. I believe that this section could be improved.

-> As suggested, the conclusion is rewritten to clarify the contribution and limitations of this study in detail and to introduce future research plans.

- References section. In general, all references are in line with MDPI standards, but in the case of references to journals, the year should be put in "bold". For example, on line 520, 2019 would be in bold. Correct all references.

->   All years in References are in bold.  

- The article has a broadly correct structure. It provides the reader with an automated analysis system that will undoubtedly be useful to the research community and, of course, to communities affected by natural disasters. I recommend that the suggested improvements be made.

-> Thanks for your positive comments on the paper

We have also updated multiple portions recommended by other reviewers. All changes are in yellow. We are confident this paper has been more academically improved. Thanks for your valuable and constructive comments on this paper.

Please stay safe and healthy, we look forward to hearing from you. 

Warm regards, 

Round 2

Reviewer 1 Report

The changes have been made, the paper seems fine now.